# Bax assembles into large ring-like structures remodeling the mitochondrial outer membrane in apoptosis

Lena Große[1,2,†], Christian A Wurm[2,†], Christian Brüser[2,†], Daniel Neumann[2], Daniel C Jans[1,2,*] & Stefan Jakobs[1,2,**]

## Abstract

The Bcl-2 family proteins Bax and Bak are essential for the execution of many apoptotic programs. During apoptosis, Bax translocates to the mitochondria and mediates the permeabilization of the outer membrane, thereby facilitating the release of pro-apoptotic proteins. Yet the mechanistic details of the Bax-induced membrane permeabilization have so far remained elusive. Here, we demonstrate that activated Bax molecules, besides forming large and compact clusters, also assemble, potentially with other proteins including Bak, into ring-like structures in the mitochondrial outer membrane. STED nanoscopy indicates that the area enclosed by a Bax ring is devoid of mitochondrial outer membrane proteins such as Tom20, Tom22, and Sam50. This strongly supports the view that the Bax rings surround an opening required for mitochondrial outer membrane permeabilization (MOMP). Even though these Bax assemblies may be necessary for MOMP, we demonstrate that at least in Drp1 knockdown cells, these assemblies are not sufficient for full cyto-chrome *c* release. Together, our super-resolution data provide direct evidence in support of large Bax-delineated pores in the mitochondrial outer membrane as being crucial for Bax-mediated MOMP in cells.

**Keywords** Bcl-2; cell death; membrane curvature; MICOS; superresolution microscopy

**Subject Categories** Autophagy & Cell Death; Membrane & Intracellular Transport; Structural Biology

The EMBO Journal (2016) 35: 402–413

See also: **G Dewson** (February 2016) and **R Salvador-Gallego** *et al* (February 2016)

## Introduction

Apoptosis, a form of programmed cell death, is a tightly regulated suicide program of eukaryotic cells (Hotchkiss *et al*, 2009). During embryonic development, it is essential for successful organogenesis, and in the adult organism, it is required to protect the organism from damaged or mutated cells that threaten normal cellular homeo-stasis (Vaux & Korsmeyer, 1999; Meier *et al*, 2000). Perturbations in its regulation contribute to numerous pathological conditions, including cancer, autoimmune, and neurodegenerative diseases (Mattson, 2000; Yuan & Yankner, 2000; Green & Kroemer, 2004).

The so-called intrinsic apoptosis pathway is triggered by several stimuli such as cytotoxic stress, DNA damage, and growth factor deprivation (Tait & Green, 2010). In this pathway, cell death is ultimately a consequence from a cascade of events in which mito-chondrial permeabilization plays a crucial role (Kroemer *et al*, 2007). In healthy cells, mitochondria sequester a potent cocktail of pro-apoptotic proteins from the cytosol, including cytochrome *c* and Smac/DIABLO. After induction of apoptosis, the integrity of the mitochondrial outer membrane is being compromised, a process called mitochondrial outer membrane permeabilization (MOMP) that results in the release of intermembrane space proteins into the cytosol. The release of cytochrome *c* initiates the activation of caspases inducing the subsequent apoptotic program (Liu *et al*, 1996; Kluck *et al*, 1997). In most cells, this event is considered as the "decision to die" and thus with MOMP a point of no return is trespassed.

In vertebrates, MOMP is controlled by the members of the Bcl-2 protein family (Martinou & Youle, 2011; Moldoveanu *et al*, 2014); key pro-apoptotic Bcl-2 proteins are the Bcl-2 proteins Bax (Bcl-2-associated X protein) and Bak (Bcl-2-antagonistic killer). Both, Bax and Bak are essential for mitochondria mediated apoptosis. In healthy cells, Bax and Bak are constantly shuttled between the mito-chondria and the cytosol. Because of different rates for the retro-translocation from the mitochondria to the cytosol, Bak resides predominantly in the mitochondrial outer membrane, whereas the

1 Department of Neurology, University Medical Center of Göttingen, Göttingen, Germany
2 Department of NanoBiophotonics, Max Planck Institute for Biophysical Chemistry, Göttingen, Germany
*Corresponding author. Tel: +49 551 2012593; E-mail: djans@gwdg.de
**Corresponding author. Tel: +49 551 2012531; E-mail: sjakobs@gwdg.de
†These authors contributed equally to this work

majority of the Bax molecules are in the cytosol (Edlich *et al*, 2011; Schellenberg *et al*, 2013; Todt *et al*, 2015).

Upon apoptosis, active Bax is not retrotranslocated, as Bax activation blocks shuttling into the cytosol and consequently Bax accumulates on the mitochondria (Wolter *et al*, 1997; Edlich *et al*, 2011). In the course of the translocation, Bax changes its conformation, oligomerizes, and inserts into the outer membrane (Hsu *et al*, 1997; Gross *et al*, 1998; Eskes *et al*, 2000). At later stages of the apoptotic process, this leads to the formation of large "Bax clusters" on the mitochondria (Nechushtan *et al*, 2001; Zhou & Chang, 2008). Although the requirement for activated Bax and Bak to induce MOMP is widely accepted, and most of the key members of the Bcl-2 family are well understood genetically, biochemically, and structurally, a major knowledge gap is the lack of mechanistic insight how Bax and Bak mediate MOMP (Tait & Green, 2010; Moldoveanu *et al*, 2014).

It has been suggested that Bax may modulate the opening of the so-called permeability transition pore complex (PTPC). The exact nature of the PTPC is still ill defined but it has been proposed to contain VDAC, ANT, and CypD (Green & Kroemer, 2004; Kroemer *et al*, 2007). However, genetic studies excluded the requirement of some of the major components of PTPC for mitochondrial-mediated apoptosis, questioning the role of the PTPC in Bax-induced MOMP (Baines *et al*, 2007).

Bax can permeabilize artificial membranes facilitating the passage of very large (2 MDa) dextran molecules (Kuwana *et al*, 2002). Substantial evidence points to the concept that Bax, alone or combined with other proteins, forms pores large enough to facilitate the release of proteins from mitochondria also in cells (Tait & Green, 2010; Renault & Manon, 2011). A caveat of the pore model, however, is that most studies have described channels only large enough to accommodate the release of cytochrome *c*, whereas in mitochondria also larger molecules are released upon induction of MOMP (Tait & Green, 2010). Alternatively, it has been suggested that activated Bax and Bak, instead of forming defined channels or pores, may lead to membrane bending and, ultimately, to the formation of lipidic pores allowing protein release (Kuwana *et al*, 2002). This model would explain the release of large proteins. Recently, electron microscopy of liposomes and purified mitochondrial outer membranes revealed large openings in the membranes that were dependent on the addition of Bax (Schafer *et al*, 2009; Gillies *et al*, 2015). However, similar opening have so far not been reported in mitochondria in cells and

it is mechanistically difficult to comprehend how a compact Bax cluster could induce the formation of such an extended opening in the mitochondria.

In this study, we demonstrate that activated Bax assembles into ring-like structures on apoptotic mitochondria. The area delineated by these rings appears to be devoid of outer membrane proteins, strongly suggesting the generation of large pores as a mechanism for outer membrane permeabilization by Bax.

## Results

### Bax forms clusters and ring-like structures of various diameters on mitochondria in apoptotic cells

The translocation of activated Bax from the cytosol to the mitochondrial outer membrane is generally accompanied by excessive mitochondrial fission and cristae remodeling (Youle & Karbowski, 2005). To visualize the localization of Bax, we refrained from co-expressing GFP-Bax fusion proteins in order to avoid potential artifacts introduced by the tag or overexpression. Instead, we induced apoptosis in human U2OS by the addition of actinomycin D in the presence of the caspase inhibitor Z-VAD-FMK and chemically fixed the cells after 18 h. At this time-point in 3–4% of all cells (*n* = 550), Bax was translocated to the mitochondria which was accompanied by mitochondrial fragmentation. These cells were regarded as apoptotic. The cells were decorated with antibodies against Bax and Tom20 or Tom22, two subunits of the translocase of the outer membrane (TOM) complex residing in the mitochondrial outer membrane. Fully in line with previous reports, we recorded numerous Bax clusters associated with fragmented mitochondria (Fig 1A). Previous studies reported that the Bax clusters associated with apoptotic mitochondria have a diameter between ~250 and ~600 nm, and contain more than 100 protein molecules (Zhou & Chang, 2008), or up to several thousand or even several tens of thousands of Bax molecules (Nechushtan *et al*, 2001). Hence these are large, compact structures resulting in very bright signals that dominate the immunofluorescence images.

STED super-resolution microscopy (or nanoscopy) enabling a lateral resolution of better than 40 nm (Hell, 2009) revealed that these Bax clusters often appeared not to be embedded into the outer membrane but to be attached to the mitochondrial surface with only a fraction of the cluster extending in the outer membrane

**Figure 1.  Bax forms ring-like structures on fragmented mitochondria during apoptosis.**

A    Left: Overlay image of a confocal recording of an apoptotic U2OS cell surrounded by non-apoptotic cells labeled with Bax (green) and Tom22 (red). Bax translocates to the mitochondria in apoptotic cells. Right: Individual display of the two color channels as shown in the overlay image. The asterisk denotes the apoptotic cell. Scale bars: 10 μm.

B, C  Dual-color image of an apoptotic U2OS cell decorated with antibodies against Bax (green) and Tom20 (red). The Bax signal was recorded in the STED mode and the Tom20 signal in the diffraction-limited confocal mode. (B, C) display the same data set. The small images are magnifications of the areas indicated by the rectangles. In (B), the color table was adjusted to the brightest Bax signal in the image resulting in a clear display of the clusters. In (C), the color table was adjusted such that the dimmer Bax rings are visible and the clusters are saturated. The arrows point to Bax rings. Scale bars: 1 μm.

D    Optical sections of an apoptotic mitochondrion labeled for Bax (green) and Tom22 (red). Shown are three optical sections and a maximum intensity projection of the entire image stack. Scale bar: 0.5 μm.

E    Bax rings on apoptotic mitochondria of various mammalian cell types. Shown are apoptotic mitochondria labeled for Bax (green) and Tom22 (red) in HT1080, SH-SY5Y, HeLa, and CV-1 cells, as indicated. Scale bars: 0.5 μm.

F    Quantification of the diameter of Bax rings in apoptotic U2OS cells. Only the long ring-axis was measured. Bax rings smaller than 200 nm exist, but were not considered.

Source data are available online for this figure.

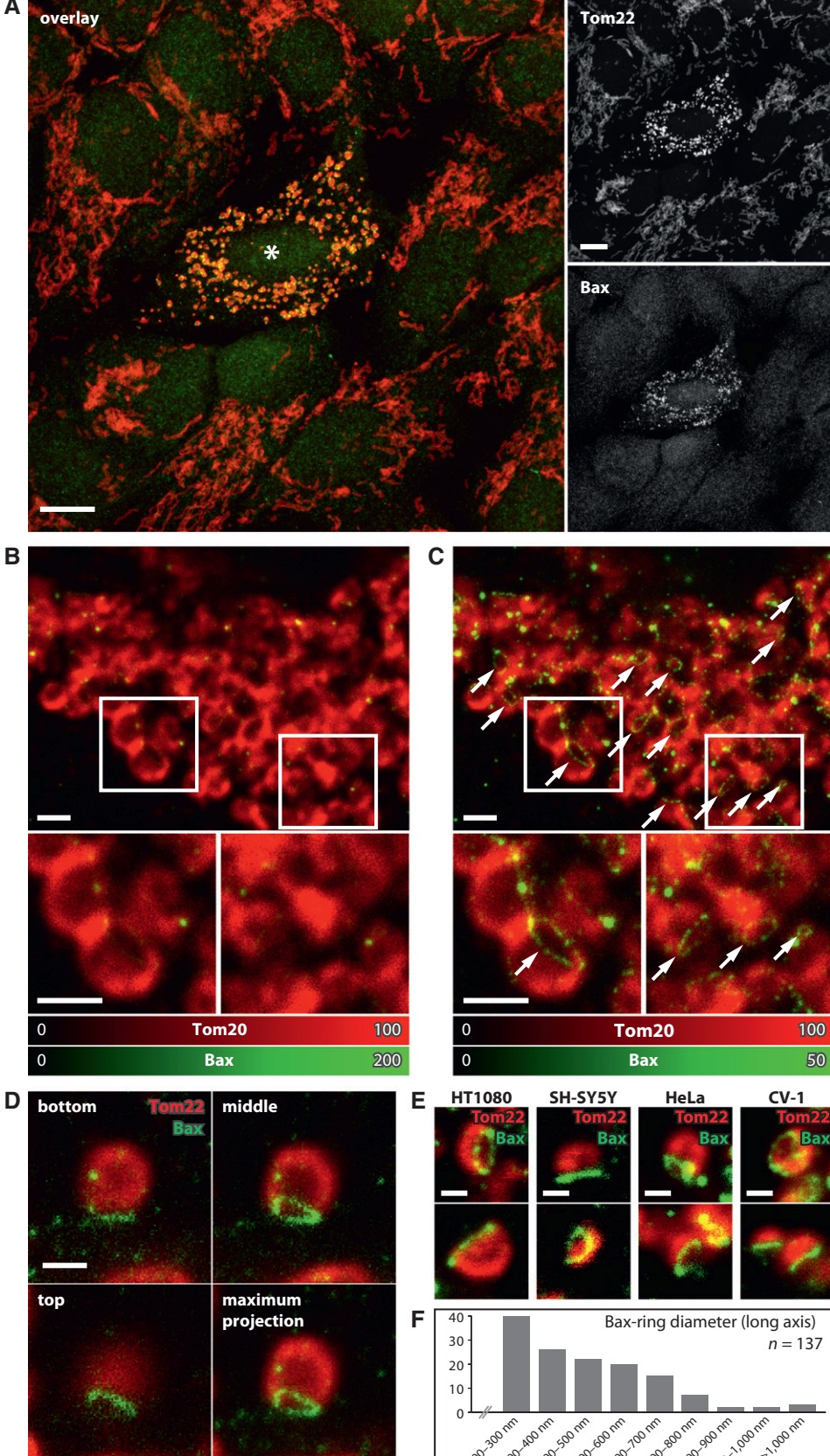

Figure 1.

(Appendix Fig S1). It is difficult to imagine how a compact Bax cluster that is attached to the mitochondrial surface could mediate MOMP.

To address the question whether these compact clusters are the only larger Bax assemblies on the apoptotic mitochondria, we scrutinized our STED super-resolution images. Detailed analysis revealed that Bax appeared to assemble not only into large clusters but also into ring-like structures, elongated and arc-shaped assemblies (Fig 1B–E). In images recorded with conventional microscopy, these unexpected Bax assemblies would be blurred because of the limited resolution. Even in the STED images these Bax assemblies were not always fully obvious, because they were relatively dim compared to the compact and bright Bax clusters (Fig 1B). Hence, we carefully adapted the imaging conditions and the color tables to visualize also the dim structures. We found that almost all Bax-positive fragmented mitochondria in U2OS cells exhibited next to Bax clusters also ring-like structures and other unexpected assemblies (Fig 1C). These novel Bax assemblies were detected with two different anti-Bax antibodies. Importantly, throughout this manuscript, we did not apply any deconvolution to the STED image data. Only raw data are shown; we merely adjusted the range of the color tables to facilitate visualization of the dim Bax assemblies (Fig 1B and C). With this approach, it is ensured that the reported Bax structures are not a result from computational artifacts due to image reconstruction.

We next asked the question whether the ring-like structures, elongated and arc-shaped assemblies represent different Bax assemblies or the same type of structure. In fact, a ring-like structure would only appear as a ring in the STED image when it was to be situated within the focal plane (Jakobs & Wurm, 2014). The impression of an elongated or arc-shaped Bax assembly might be due to a ring-like structure that is in line with the viewing direction of the microscope. To address this issue, we recorded 3D stacks of arcs and elongated assemblies using STED nanoscopy (Fig 1D). All Bax assemblies analyzed ($n > 20$) that appeared as an elongated structure in the 2D STED images proved to be ring-like structures in the 3D recordings. Although we have investigated only a limited set of elongated and arc-shaped assemblies, and hence cannot exclude occasional elongated Bax structures, our findings suggest that the majority of the elongated and arc-shaped assemblies represent ring-like Bax arrangements that are integrated into the mitochondrial outer membrane. For simplicity, we will refer to these Bax ring-like structures in the rest of this manuscript as Bax rings.

To determine whether such Bax rings are a general phenomenon occurring during apoptosis, we analyzed four other mammalian cell lines, namely HT1080, SH-SY5Y, HeLa, and CV-1 cells. In all four cell lines, we observed Bax rings on apoptotic mitochondria, strongly suggesting that these rings are a common structure in apoptotic cells (Fig 1E).

The majority of the Bax rings were practically not resolvable by conventional light microscopy. STED nanoscopy demonstrated that their sizes varied considerably (Fig 1F). Most rings were smaller than 400 nm in diameter along the long ring-axis; we note, however, that many rings were too small to be resolved even by STED nanoscopy. Hence, there appears to be a continuum of ring diameters without a clearly defined final size. Frequently, the labeling of the rings appeared to be speckled. This might be due to imperfect antibody decoration, or it might indicate that these assemblies are composed of different proteins.

Importantly, we noticed that in dual-color images of cells decorated with antibodies against Tom20 and Bax, the area surrounded by the Bax rings was often devoid of a Tom20 signal (Fig 1C). To verify that the membrane area enclosed by the Bax ring is indeed

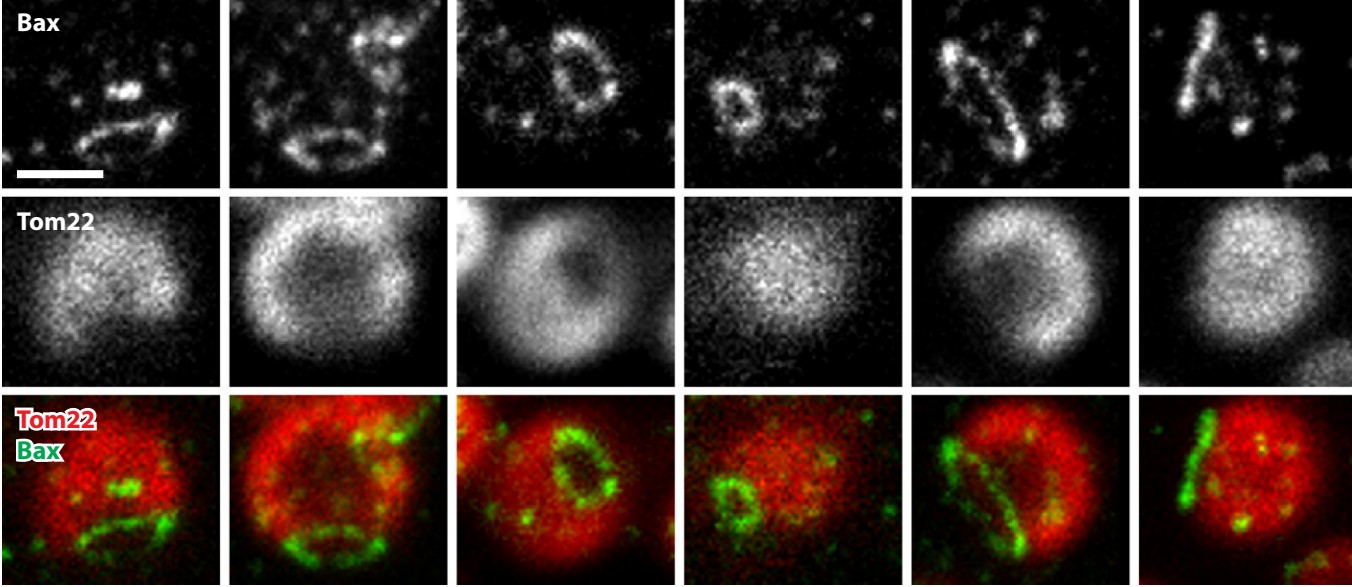

**Figure 2. The area enclosed by a Bax ring is devoid of the mitochondrial outer membrane protein Tom22.**

Apoptotic mitochondria were decorated with antibodies against Bax (green) and Tom22 (red). The Bax signal was recorded in the STED mode. Shown are the individual color channels as well as an overlay (from top to bottom) of several apoptotic mitochondria. We used Lagrange interpolation to double the number of pixels. Note that throughout this manuscript except for contrast stretching, no image manipulation was applied. Scale bar: 0.5 μm.

Source data are available online for this figure.

devoid of subunits of the TOM complex, we re-evaluated 3D recordings of apoptotic mitochondria decorated with an antibody against Tom22. Focusing through a mitochondrion clearly showed lack of Tom22 in the area delineated by the Bax ring (Fig 1D). To substantiate this finding further, we recorded numerous Bax rings in different orientations on the mitochondria (Fig 2). Depending on the positioning of the ring, the lack of Tom22 signal within the ring was more or less visible. Although our data cannot exclude the existence of Bax rings not exhibiting this phenomenon, the majority of the Bax rings showed a marked decrease in or the absence of Tom20 and Tom22 signal in their interior. To demonstrate that this phenomenon is not restricted to subunits of the TOM complex, we decorated apoptotic cells with antibodies against Bax and Sam50, an integral component of the mitochondrial outer membrane sorting and assembly machinery (SAM or TOB) complex (Appendix Fig S2A). As in apoptotic mitochondria decorated for Tom20 or Tom22, we found that the interior of the Bax rings was largely devoid of a signal for Sam50, strongly suggesting that the interior of the outer membrane embedded Bax rings is indeed devoid of membrane proteins.

So far, we used the caspase inhibitor Z-VAD-FMK to prevent the apoptotic cells from detaching from the cover class. To verify that Z-VAD-FMK does not induce Bax ring formation, we also imaged cells grown in the absence of this inhibitor. Upon induction of apoptosis, we observed Bax rings, demonstrating that the rings form also in the absence of Z-VAD-FMK (Appendix Fig S2B). However, we found only relatively few apoptotic cells under these conditions. We assume that the assembly of Bax rings and the detachment of cells are tightly linked. Hence, presumably, there is only a very short time span to observe Bax rings in the absence of Z-VAD-FMK.

Together, we have shown using STED nanoscopy that upon induction of apoptosis, Bax is translocated to the mitochondrial outer membrane. Next to large clusters, Bax assembles into rings of varying sizes, which appear to be integrated into the outer membrane. Intriguingly, the interior of these Bax rings appeared to be largely devoid of outer membrane marker proteins including Tom20, Tom22, and Sam50. This observation shows that the Bax rings remodel the delineated outer membrane.

## The formation of Bax rings is strictly correlated to cytochrome *c* release in apoptotic wild-type cells

To address the question whether the formation of Bax rings is correlated to MOMP, we induced apoptosis in human U2OS cells with actinomycin D and decorated the cells with antibodies against Tom22, Bax, and cytochrome *c* (Fig 3). In non-apoptotic cells, the tubular mitochondria contained cytochrome *c* and Bax had not translocated to the mitochondria (Fig 3A). In apoptotic cells, the mitochondria were fragmented and had released cytochrome *c* (Fig 3B). When inspecting these images for ring-like structures, we noticed that all mitochondria analyzed that carried a recognizable ring-like structure (*n* > 200) had released cytochrome *c*. Not all mitochondria that had released cytochrome *c* carried a visible Bax ring, but this does not evidence the absence of a Bax ring, because the ring might have been in a different focal plane. Together, we have not been able to identify a wild-type mitochondrion with a Bax ring that had not released cytochrome *c*.

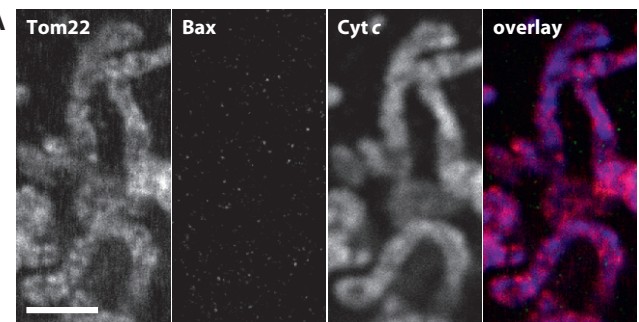

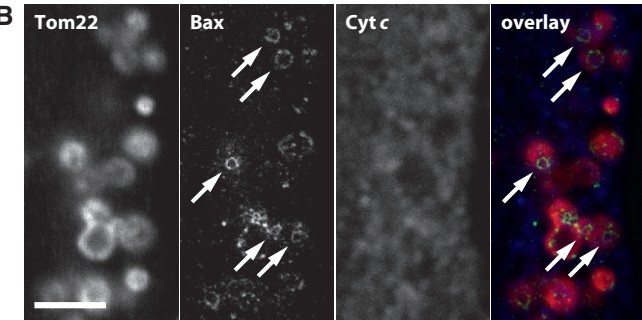

**Figure 3. The formation of Bax rings is strictly correlated to cytochrome *c* release in apoptotic wild-type cells.**

A, B    Three-color images of mitochondria of unchallenged (A) and apoptotic (B) U2OS cells. From left to right: Tom22, Bax (recorded in the STED mode), cytochrome *c*, and an overlay of all three channels. The arrows denote Bax rings. Scale bars: 2 μm.

Source data are available online for this figure.

## Bax rings are not sufficient for cytochrome *c* release in Drp1 knockdown cells

In wild-type U2OS cells, Bax rings appear to be involved in the release of cytochrome *c*. To address the question whether Bax rings are sufficient for full cytochrome *c* release, we made use of the observation that knockdown of the mitochondrial division protein Drp1 prevents mitochondrial fragmentation and leads to a delay of cytochrome *c* release upon stimulation of apoptosis. The release of Smac/DIABLO is not altered by the absence of Drp1 (Parone *et al*, 2006; Estaquier & Arnoult, 2007). Unchallenged U2OS cells with reduced levels of Drp1 exhibited a hyperfused mitochondrial network with long mitochondrial tubules. Upon induction of apoptosis with actinomycin D, the mitochondrial network had changed its shape and next to elongated mitochondria, swollen mitochondria, and aggregated fragmented mitochondria appeared (Fig 4A). This mitochondrial shape change was accompanied by the translocation of Bax to the mitochondria. In line with previous studies, we found that only a subset of the Bax-positive mitochondria had fully released cytochrome *c* (Fig 4A), but the majority (> 95%) had released Smac/DIABLO (Appendix Fig S3) (Parone *et al*, 2006; Estaquier & Arnoult, 2007).

STED nanoscopy revealed that the mitochondria in apoptotic Drp1 knockdown cells exhibited in addition to the Bax clusters also Bax rings (Fig 4B). We note, however, that identifying Bax rings in Drp1 knockdown cells was substantially more difficult than in apoptotic wild-type cells. This might indicate that apoptotic Drp1

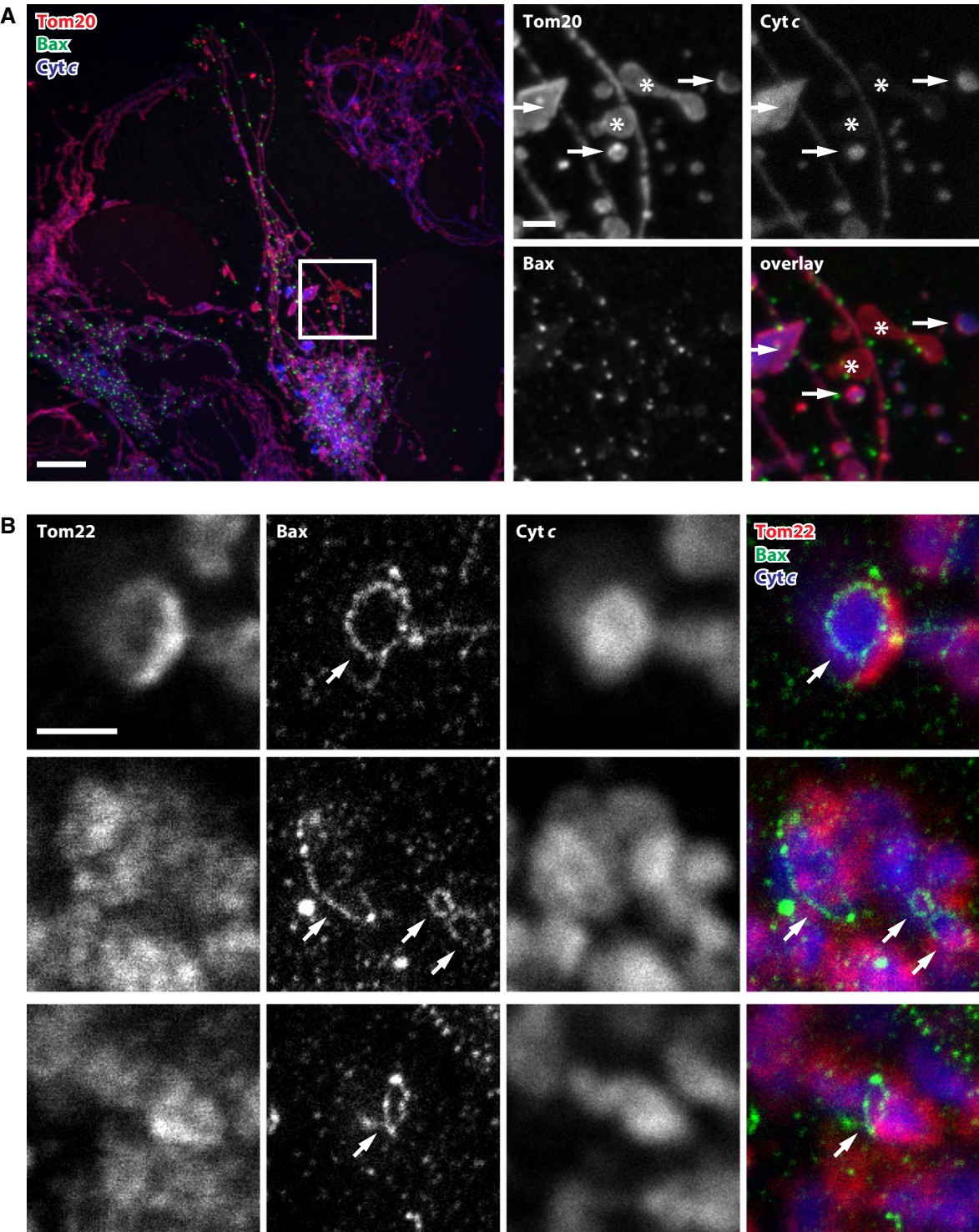

**Figure 4.  In apoptotic Drp1 knockdown cells, the formation of Bax rings is not sufficient for full cytochrome *c* release.**

A   In apoptotic Drp1 knockdown U2OS cells, cytochrome *c* release is delayed. Confocal overview image of an apoptotic Drp1 knockdown cell labeled with antibodies against Tom20 (red), cytochrome *c* (blue), and Bax (green). Large image: Overlay of all three channels. Small images: Magnifications of the area indicated by the rectangle in the large image. Shown are the individual channels, as indicated. The arrows point to mitochondria that had not released cytochrome *c*; the asterisks denote mitochondria that had released cytochrome *c*. Scale bars: 10 µm (overview) and 2 µm (magnifications).

B   In Drp1 knockdown cells, Bax-ring formation and cytochrome *c* release is not correlated. Three-color images of apoptotic mitochondria of Drp1 knockdown cells. From left to right: Tom22, Bax (recorded in the STED mode), cytochrome *c*, and an overlay of all three channels. The arrows point to Bax rings. Scale bar: 1 µm.

Source data are available online for this figure.

knockdown cells indeed exhibit fewer rings, that the rings are less developed, or that because of the architecture of the mitochondria in Drp1 knockdown cells, they are more difficult to visualize. Interestingly, and in line with our observation of fewer Bax rings on Drp1-deficient mitochondria, a previous study had shown that Drp1 stimulates the oligomerization of Bax, probably by the formation of Drp1-induced membrane hemifusion intermediates (Montessuit *et al*, 2010). We observed on apoptotic mitochondria of Drp1 knockdown cells elongated and arc-shaped Bax assemblies that appeared not to form a closed ring. The STED data showed that discernible Bax rings were preferentially established on fragmented mitochondria. On elongated mitochondria, the Bax rings were very rare, and, if any, their diameter was very small. Importantly, we observed in Drp1 knockdown cells fragmented mitochondria exhibiting clear Bax rings that had not fully released cytochrome *c*, a scenario we never observed in apoptotic wild-type cells (Fig 4B).

We conclude that in Drp1 knockdown cells Bax translocates to the mitochondria, forming both clusters as well as occasional rings and potentially other assemblies. The majority of these Bax-positive mitochondria had released Smac/DIABLO, whereas the release of cytochrome *c* was delayed. We identified numerous mitochondria exhibiting Bax rings that had not fully released cytochrome *c*, evidencing that in Drp1 knockdown cells these Bax assemblies are not sufficient for the full release of cytochrome *c*. This delays but does not inhibit the further progression of apoptosis (Estaquier & Arnoult, 2007).

### The redistribution of the cristae junction-associated MICOS complex precedes the release of cytochrome *c* in apoptotic Drp1 knockdown cells

The existence of mitochondria in Drp1 knockdown cells that exhibited a Bax ring but had not released their entire cytochrome *c* content supported the view that in addition to Bax assembly, another mechanism may control cytochrome *c* release. Mitochondrial cristae are involutions of the inner membrane and it has been estimated that most cytochrome *c* (> 80%) (Scorrano *et al*, 2002) is sequestered in the intermembrane space enveloped by the cristae membranes. The cristae membranes are connected by cristae junctions, relatively uniform tubular structures of typically 20–50 nm diameter, with the inner boundary membrane. Because of their architecture, the cristae junctions have been suggested to be involved in controlling the release of proteins from the cristae lumen (Scorrano *et al*, 2002).

The recently discovered multisubunit MICOS complex (mitochondrial contact site and cristae organizing system) is required for the maintenance of cristae junctions (Harner *et al*, 2011; Hoppins *et al*, 2011; von der Malsburg *et al*, 2011; Alkhaja *et al*, 2012). For HeLa cells depleted of Mic60 (mitofilin), a core component of the MICOS complex, it was shown that their mitochondria released cytochrome *c* faster than wild-type cells upon induction of apoptosis, suggesting that the MICOS complex may have a role in controlling cytochrome *c* release (Yang *et al*, 2012).

In STED images of healthy cells, Mic60 is seen as punctuate structures reflecting MICOS clusters at individual cristae junctions (Appendix Fig S4) (Jans *et al*, 2013). We reasoned that MICOS might change its inner-mitochondrial localization during apoptosis to modify the architecture of the cristae junctions. Indeed, we found

that in apoptotic wild-type cells, Mic60 no longer displays a distinct punctuate localization, but is rather uniformly distributed over the mitochondria, suggesting a redistribution of MICOS during apoptosis. To address the question whether this redistribution occurs before or after cytochrome *c* release, we relied on apoptotic Drp1 knockdown cells where cytochrome *c* release is delayed. In order to analyze the MICOS distributions objectively from a larger number of mitochondria, we analyzed the normalized local variance of the fluorescence signal within the labeled mitochondria (Wurm *et al*, 2011). Generally, dense clustering results in lower variance values, whereas sparse clustering results in higher values.

Using this approach, we compared the distribution of the two MICOS core components Mic27 and Mic60 in unchallenged and apoptotic mitochondria of Drp1 knockdown cells (Fig 5). As in wild-type cells, we observed a strong change in the Mic27 and the Mic60 distribution after translocation of Bax. The normalized local variance values of the fluorescence signal decreased, reflecting the decrease in discernible Mic27 and Mic60 clusters (Appendix Fig S4). However, when comparing Bax-positive mitochondria that had released cytochrome *c* with those that had not,

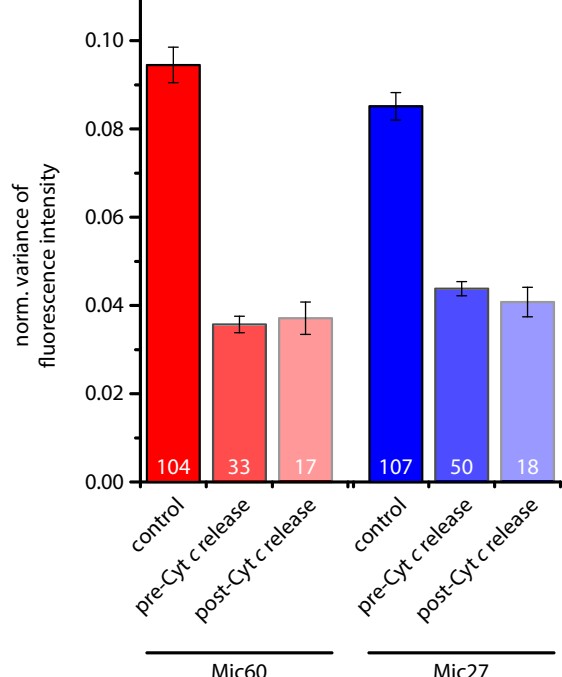

**Figure 5. The redistribution of the MICOS core components Mic27 and Mic60 precedes the release of cytochrome *c* in apoptotic Drp1 knockdown cells.**

Drp1 knockdown cells were treated with DMSO (control) or with actinomycin D for 14 h to induce apoptosis and were decorated with antisera against Bax, cytochrome *c*, and Mic27 or Mic60 (see Appendix Fig S4). Three-color images were taken, whereby Mic27 or Mic60 was recorded in the STED mode. Relying on the Bax and the cytochrome *c* signals, we discriminated between healthy and apoptotic mitochondria before and after cytochrome *c* release. The distributions of Mic27 or Mic60 were analyzed by determining the normalized variance of the fluorescence intensity, which is a sensitive measure for the distribution of the labeled protein. The numbers within the columns represent the numbers of images analyzed and the error bars represent the standard error of the mean.

we observed no further difference in the normalized fluorescence variance values. This indicated that the actual cytochrome *c* release is independent of a change in the localization of Mic27 and Mic60.

Our findings demonstrate that MICOS loses its distinct localization into individual clusters early in the apoptotic process. The disintegration of the ordered MICOS localization precedes the release of cytochrome *c* in Drp1 knockdown cells. This finding suggests that the MICOS complex, although it is required for the maintenance of cristae junctions, is not immediately involved in the actual release of cytochrome *c*.

## Discussion

Translocation of the pro-apoptotic protein Bax to the mitochondrial outer membrane in response to many apoptotic stimuli is essential for MOMP and the subsequent release of cytochrome *c* ultimately results in cell death. However, the mechanistic means by which Bax induces membrane permeabilization remain elusive (reviewed in: Tait & Green, 2010; Martinou & Youle, 2011; Renault & Manon, 2011).

In this study, we report that activated Bax assembles into ring-like structures in the outer membrane of apoptotic mitochondria. In apoptotic human U2OS cells as well as in other apoptotic cell lines, we observed such Bax rings on the majority of the fragmented mitochondria, so these are frequent assemblies. Although some rings may be larger than 500 nm in diameter, the majority is smaller than 400 nm, making it challenging or even impossible to distinguish a ring-like assembly from a cluster when relying on diffraction-limited conventional light microscopy instead of diffraction-unlimited nanoscopy. Another reason that these structures have not been reported earlier may be due to the fact that the high fluorescence signals of the compact clusters of activated Bax can dominate an image, so that the rather dim Bax rings may easily disappear in the background of the image and are overlooked. In addition to smoothly labeled Bax rings, we repeatedly observed also Bax rings that were non-homogeneously labeled. This might indicate that at least a subset of the Bax rings are assemblies of Bax with another protein, potentially Bak. Indeed, activated Bax has been reported to interact with Bak into complexes involved in MOMP (Nechushtan *et al*, 2001; Zhou & Chang, 2008). Hence our data would be in line with ring-like structures consisting of Bax, Bak, and potentially other proteins.

Although Bax rings are frequent on apoptotic wild-type mitochondria, we rarely recorded fully established Bax rings on Drp1 knockdown cells amidst Bax clusters and other Bax assemblies such as arcs and elongated structures. This is somewhat surprising, since most of the Bax-positive mitochondria in Drp1 knockdown cells had released Smac/DIABLO, indicative of a permeabilization of the outer membrane. This finding suggests that, at least in the Drp1 knockdown cells, also other Bax assemblies than Bax rings are sufficient for the permeabilization of the outer membrane. As reported previously (Parone *et al*, 2006; Estaquier & Arnoult, 2007), we found that in the Drp1 knockdown cells Smac/DIABLO release is not delayed, whereas cytochrome *c* release is. Thus, the data further supports the concept that in addition to Bax-induced MOMP, a structural remodeling of the inner-mitochondrial membrane is occurring to facilitate cytochrome *c* release (Scorrano

*et al*, 2002). In agreement with this hypothesis, the intermembrane space proteins optic atrophy protein 1 (OPA1) and presenilin-associated rhomboid-like protein (PARL) have been shown to influence cristae or cristae junctions remodeling during apoptosis (Cipolat *et al*, 2006; Frezza *et al*, 2006; Yamaguchi *et al*, 2008). In this study, we demonstrate that the MICOS complex required for cristae junction maintenance changes its inner-mitochondrial distribution before full cytochrome *c* release in Drp1 knockdown cells, rendering it rather unlikely that this protein complex is involved immediately in the release of cytochrome *c*. Hence together, these data suggest a complex succession of events that include MOMP, and a re-organization of inner-membrane proteins before and upon cytochrome *c* release.

Most studies investigating the molecular mechanisms mediating MOMP favor the concept that Bax, alone or in combination with other proteins, forms either proteinaceous channels or lipidic pores *in vivo*. Indeed, in artificial membranes, Bax has been shown to form proteinaceous channels (Antonsson *et al*, 1997; Martinez-Caballero *et al*, 2009). A caveat to this model is that MOMP facilitates in cells the release of proteins substantially larger than cytochrome *c*, whereas most studies described only proteinaceous Bax channels large enough to accommodate cytochrome *c* (Tait & Green, 2010).

The concept of the formation of extended lipid pores in the membrane facilitated by the interaction of Bax and Bak with outer membrane lipids accounts for the release of large intermembrane space proteins. This concept got recently support by cryo-electron microscopy of liposomes and vesicles generated of mitochondrial outer membrane (OMVs) that were permeabilized by the addition of Bax. The liposomes and the OMVs revealed large round openings with a diameter of ~25–100 nm and of ~100–160 nm, respectively, with some openings as large as 300 nm. The diameter of these openings is consistent with the ability of Bax to mediate the passage of dextran molecules as large as 2 MDa (Kuwana *et al*, 2002). In the cryo-electron images, these openings had relatively smooth edges, which is suggestive of Bax-induced lipidic pores. Unfortunately, the distribution of Bax in these vesicles is not known and similar pores have so far not been reported in mitochondria. Intriguingly, the size of the openings in the isolated vesicles is consistent with the size of the Bax rings we observed on the mitochondria of apoptotic human cells.

Hence, it is tempting to assume that the Bax-induced openings in the liposomes and the Bax rings on mitochondria described in this study relate to the same type of structures. In this case, the Bax rings would be predicted to delineate an opening in the outer membrane and a partial or full absence of outer membrane within the rings would be expected. Indeed, our STED images of mitochondria of apoptotic wild-type cells decorated with antisera against Bax and the abundant outer membrane proteins Tom20, Tom22, or Sam50 support this assumption (Figs 1C–E and 2; Appendix Fig S2). To further substantiate this observation, Fig 6 shows seven optical sections through a mitochondrion labeled for Bax and Tom22. Tom22 is homogenously distributed over the mitochondrial outer membrane except in the region delineated by the Bax ring (Fig 6). This distribution is fully in line with a Bax ring-mediated opening in the outer membrane.

Similar conclusions were drawn from an independently performed study, which recently came to our attention (Salvador-Gallego *et al*, 2016). Deletion of the C-terminus of Bax abolishes

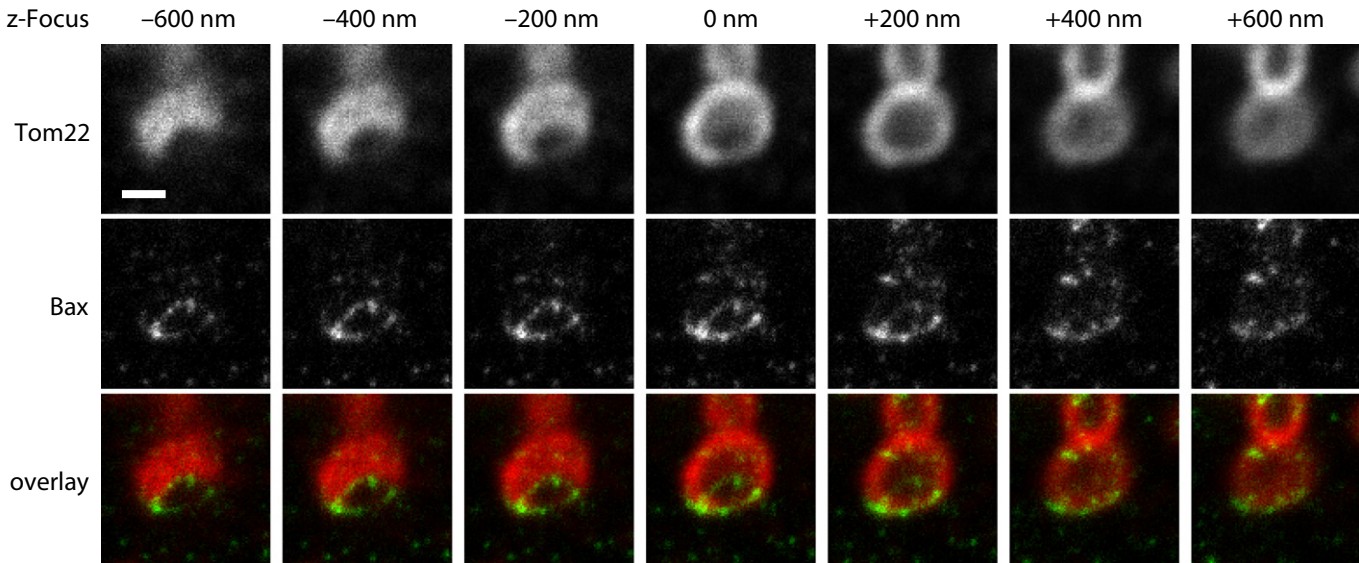

**Figure 6. 3D recordings of apoptotic mitochondria suggest the formation of a Bax-ring-induced pore in the mitochondrial outer membrane.**
Shown are seven optical sections through a mitochondrion of an apoptotic U2OS cell. Top and middle row: Tom22 and Bax (recorded in STED) channel, respectively. Bottom row: overlay (Tom22: red; Bax: green). Note the absence of Tom22 signal in the area enclosed by the Bax ring. Displayed are raw data. Scale bar: 0.5 μm.

Source data are available online for this figure.

its mitochondrial localization and its pro-apoptotic activity (Nechushtan *et al*, 1999; Schinzel *et al*, 2004). Also, apoptosis is largely suppressed in cells expressing Bax mutants that localize to mitochondria but are affected in their ability to homo-oligomerize (Peng *et al*, 2013). Some Bax mutants resulting in a compromised Bax retrotranslocation lead to an accumulation of Bax on the mitochondrial outer membrane, but require further stimulation to facilitate apoptosis (Todt *et al*, 2013, 2015). We propose that the investigation of cells expressing such mutants will help to further dissect the function of the Bax rings in the succession of events leading to cytochrome *c* release.

Together, we have shown using STED nanoscopy that upon induction of apoptosis, Bax translocates to the mitochondria to form, alone or possibly with other proteins, next to large and compact clusters also rings in the mitochondrial outer membrane. The area enclosed by the Bax rings is largely devoid of the outer membrane proteins Tom20, Tom22, and Sam50, strongly supporting the view that the Bax rings delineate a pore required for MOMP.

# Materials and Methods

### Cell culture and transfection

Cultured U2OS, HT1080, SH-SY5Y, HeLa, or CV-1 cells were grown in DMEM with glutamax and 4.5% (w/v) glucose (Invitrogen, Carlsbad, CA, USA) supplemented with 10,000 U/ml penicillin, 10,000 μg/ml streptomycin (Merck Millipore, Darmstadt, Germany), 100 nM Na pyruvate (Sigma-Aldrich, St. Louis, MO, USA), and 10% (v/v) FCS (Merck Millipore, Darmstadt, Germany) at 37°C and 5% $CO_2$.

For Drp1 knockdown, the Drp1-shRNA-expressing plasmid pREP4 (Lee *et al*, 2004) was used. Plasmids were transiently transfected by electroporation using the nucleofector Kit V (Lonza, Basel,

Switzerland). To enrich transfected cells, selection was performed 24 h after transfection using 200 μM hygromycin B (Life Technologies, Carlsbad, USA) for 2 days, followed by 5–6 days of incubation with lower amounts of hygromycin B (50 μM) in the growth medium.

### Induction of apoptosis

Apoptosis was induced with 10 μM actinomycin D (Merck Millipore, Darmstadt, Germany) and 20 μM z-vad-fmk (Santa Cruz Biotech, Dallas, USA) in U2OS (18 h), in HT1080 (18 h), SH-SY5Y (24 h), HeLa (24 h), and CV-1 (24 h) cells. In Drp1 knockdown cells, apoptosis was induced with 10 μM actinomycin D (without z-vad-fmk) for 18 h.

### Sample preparation for STED microscopy

Samples were essentially prepared as described previously (Wurm *et al*, 2010). In brief, cells were fixed with 8% (w/v) formaldehyde in PBS (137 mM NaCl, 2.68 mM KCl, 10 mM $Na_2HPO_4$, pH 7.4) for 5 min at 37°C, extracted with 0.5% (v/v) Triton X-100 in PBS, blocked with 10% (w/v) bovine serum albumin (BSA) in PBS, and incubated with diluted primary polyclonal antibodies against Bax (Merck Millipore, Darmstadt, Germany), Mic60 (Abcam, Cambridge, England), Mic27 (Atlas, Stockholm, Sweden), Smac/DIABLO (Abcam, Cambridge, MA, USA), Tom20 (Santa Cruz Biotechnology, Santa Cruz, CA, USA), or incubated with diluted primary monoclonal antibodies against Bax (Trevigen, Gaithersburg, MD, USA), cytochrome *c* (BD Bioscience, Heidelberg, Germany) for 1 h. After three washing steps in PBS, incubation in 0.5% (v/v) Triton X-100 in PBS, and blocking in 10% (w/v) BSA in PBS, the primary antibodies were detected with secondary antibodies against mouse or rabbit (goat anti-rabbit or sheep anti-mouse Alexa Fluor 488 (Invitrogen,

Carlsbad, USA) and sheep anti-mouse or goat anti-rabbit antibodies (Jackson Immuno Research Laboratories, West Grove, PA, USA) custom-labeled with KK114 (Kolmakov *et al*, 2010) or Alexa 594 (Life Technologies, Carlsbad, USA) for 1 h.

In order to introduce a third immunolabel using a second primary mouse antibody, samples prepared as described above were further treated with detergent (0.5% (v/v) Triton X-100 in PBS) and 10% (w/v) BSA in PBS. Subsequently, free binding sites of the secondary anti-mouse antibodies were blocked with mouse anti-chicken antibodies (Biozol, Eching, Germany) in 10% (w/v) BSA in PBS for 1 h. After a fixation step with 8% (w/v) formaldehyde in PBS for 5 min, extraction with 0.5% (v/v) Triton X-100 in PBS, and blocking with 10% (w/v) BSA in PBS, the samples were incubated with cytochrome *c* antibodies directly coupled to ATTO532 (Life Technologies, Carlsbad, USA) or anti-human Tom22-AbberiorSTAR488 (Miltenyi Biotec, Bergisch Gladbach, Germany) antibodies. Finally, the samples were washed in PBS and mounted in Mowiol with 0.1% (w/v) DABCO (Sigma-Aldrich, St. Louis, MO, USA). All Tom22 immunolabelings shown in the manuscript are three-color stainings, although not all channels are necessarily displayed.

### Determination of the normalized variance of the fluorescence intensity

The normalized variance of the fluorescence intensity of STED images was determined essentially as described previously (Wurm *et al*, 2011). In brief, first for each individual pixel within the raw STED image the local variance of the fluorescence intensity around the respective pixel was determined in round regions of interest (ROIs) with a diameter of 7 pixels (~140 nm). To evaluate the results of the analysis independent from the absolute brightness of the structures, the individual variance values were normalized to the squared average fluorescence intensity of the respective ROI, giving the normalized variance values. The variance calculation was repeated using each pixel successively as an ROI center, resulting in an image in which each pixel represents the local normalized variance. The mitochondria-containing fraction of the image was selected by image segmentation using masks. For each image, the average normalized variance of the mitochondria-containing fraction was calculated.

### Confocal microscopy

Confocal microscopy was performed with a beam scanning confocal microscope (TCS SP5, Leica or TCS SP8, Leica).

### STED microscopy

Images for quantification of protein distributions before and during apoptosis were acquired using a custom-built STED setup (Wurm *et al*, 2011). Dual-color STED microscopy was performed using a 2-color Abberior STED 775 QUAD scanning nanoscope (Abberior Instruments) equipped with a Katana-08 HP laser (Onfive GmbH, Regensdorf, Switzerland). In brief, the fluorophores Alexa594 and KK114 were excited at 594 and 640 nm, respectively, and STED was performed at 775 nm for both color channels. The dye AlexaFluor488 was excited at 488 nm and recorded confocally. All images are raw

data. Except for contrast stretching and Lagrange interpolation to double the pixel number in Figs 1E and 2 and Appendix Fig S2, no image processing was applied.

**Expanded View** for this article is available online.

### Acknowledgements

We thank Richard J. Youle, NIH, for kindly providing the plasmid pREP4-U6-shRNA-DRP1. We thank Jaydev Jethwa for carefully reading the manuscript. This work was supported by the Cluster of Excellence and DFG Research Center Nanoscale Microscopy and Molecular Physiology of the Brain (to SJ).

### Author contributions

LG, CAW, CB, DN, and DCJ performed research; LG, CAW, CB, DCJ, and SJ analyzed data; DCJ and SJ designed the study and coordinated the experimental work; SJ wrote the manuscript with contributions from all authors.

### Conflict of interest

The authors declare that they have no conflict of interest.

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
