## [Review Process File · The EMBO Journal]

Manuscript EMBO-2015-92789

Bax assembles into large ring-like structures remodeling the mitochondrial outer membrane in apoptosis

Lena Große, Christian Wurm, Christian Brüser, Daniel Neumann, Daniel Jans and Stefan Jakobs

Corresponding author: Stefan Jakobs, Max-Planck-Institute for Biophysical Chemistry

Review timeline:

Submission date:	07 August 2015
Editorial Decision:	15 September 2015
Revision received:	13 October 2015
Editorial Decision:	09 November 2015
Additional Correspondence:	13 November 2015
Revision received:	23 November 2015
Accepted:	01 December 2015

Editor: Andrea Leibfried

Transaction Report:

1st Editorial Decision

15 September 2015

Thank you for submitting your manuscript for consideration by the EMBO Journal. It has now been seen by three referees whose comments are shown below.

As you will see, the referees appreciate your analyses and provide constructive input. They think, however, that additional data and information are needed to better support your conclusions and to make your manuscript a good candidate for publication in The EMBO Journal. Most importantly, referee #1 requests that you generalize your findings (point 1). I asked the referees to cross-comment on each other, and referee #3 agreed that addressing point 1 of referee #1 would be required to strengthen your work.

I would thus like to invite you to submit a revised version of the manuscript, addressing the comments of all three reviewers. I should add that it is EMBO Journal policy to allow only a single round of revision. In the interest of time, it would be also good if you could briefly outline to me how you can address the other issues raised by the referees. I think some of the points can be addressed by changing the text, but others would need further experiments. You can either call me or reply to this message to provide this outline and to discuss the revision.

Thank you for the opportunity to consider your work for publication. I look forward to your revision.

REFeree REPORTS

Referee #1:

Defining the nature and structure of the enigmatic apoptotic pore is one of the most important questions in the field. The current manuscript describes STED super resolution microscopy to attempt to image the putative apoptotic pore. Using immunohistochemistry of Bax in U2OS cells, the authors report the detection of large, bright clusters but also ring structures the lumen of which is devoid of mitochondrial membrane proteins. They also report that the formation of these rings is not sufficient for cytochrome c release in the absence of Drp1-dependent mitochondrial remodelling. The manuscript is very clearly written and the data for the most part convincing and objectively interpreted.

General comments

1. In order to ascertain the generality of their findings, their studies should be extended to multiple cell lines. Also to rule out the possibility that the structures, particularly the clusters are not an artefact of formaldehyde fixation it should be complemented with GFP-Bax expressed in Bak/Bax null cells.
2. In Drp1 k/d cells, the authors report other arc-like structures. These could represent partially formed pores or rings comprising other mitochondrial proteins in particular Bak. This should be addressed by examining Bak^{-/-} cells (MEFs or HCT116 for example). Also can the authors show that 100% of Bax is labelled with antibody?
3. The authors conclude that the clusters are "loosely attached" to mitochondria. This cannot be determined even with super resolution microscopy and should be interpreted more cautiously.
4. The authors correlate the formation of the ring structure with MOMP in apoptotic cells. What percentage of the cells treated with actD at this timepoint were apoptotic? How were individual cells determined to be apoptotic? Also this aspect of the study would be significantly strengthened by imaging mutants of Bax that do not oligomerise and/or mediate MOMP.
5. That Smac is released indicates that MOMP has occurred but suggests that Drp1 is required for full cyt c release from cristae. These cells will likely still die and undergo cytochrome c just not as quickly. So what happens at later timepoints? The statement should be more explicit to state that these assemblies are not sufficient for FULL cyt c release.

Specific points

1. Abstract should read, "The Bcl-2 family proteins Bax AND BAK ARE essential..."
2. The Ow et al 2008 review should be replaced with the original research article from Xiaodong Wang's group.
3. The Introduction discusses Bax translocation to mitochondria during apoptosis. To more accurately reflect current understanding the new phenomenon of "retrotranslocation" should be discussed.
4. The sentence, "A careful adaptation of the imaging conditions and the color tables demonstrates that almost all Bax-positive fragmented mitochondria in U2OS cells exhibit next to Bax clusters also these unexpected assemblies (Fig. 1C)" should be reworded to clarify.

Referee #2:

The presented study by Grosse et al used super-resolution STED imaging to investigate how specific proteins, involved in permeabilization of mitochondrial outer membranes, form ring-like structures upon induction of apoptosis. The study also gives indications that the ring-like structures surround holes in the membranes, through which larger proteins can be released. The manuscript is well written, the presented results and conclusions are interesting and seem solid and are likely to provide important clues to the understanding of mitochondrial outer membrane permeabilization.

I have no essential suggestions for revisions of this manuscript. Apart from some minor matters, it can in my view be published in its present form.

Minor points:

p.14, second section: Can the authors give some more information on how the ROIs for the variance determination were selected?

p.14-15, STED microscopy: Please add information on what excitation wavelengths that were used. Was a 775nm laser used for the STED beam, and does then that mean Alexa 488 labeled targets were only confocally recorded?

Referee #3:

Comments on Manuscript by Große et al.

Using super-resolution STED microscopy, Große et al. have identified a novel ring-shaped structure which is formed during apoptosis. These rings are composed of the mitochondrial outer membrane permeabilization (MOMP)-inducing protein Bax. Furthermore, the area enclosed by the rings seems to be devoid of the mitochondrial outer membrane proteins Tom20 and Tom22, which, according to the authors, would support that idea that Bax forms lipidic pores. Using cells in which the membrane fission protein Drp1 is downregulated, they demonstrate that Bax ring assemblies are not sufficient for the release of cytochrome c. The cristae junction organizing complex MICOS seems to be disassembled after Bax activation and before cytochrome c release.

MOMP is considered as the 'point of no return' during apoptosis, and the events leading to this state have been studied extensively in recent years. It remains to be established however, how Bax oligomerization causes the formation of pores in the outer membrane, and what is the exact nature of these complexes. Formation of the Bax ring structures reported by the authors is thus an extremely interesting observation because it could be the first visualization of the Bax-induced pore in cells, and even though there is little mechanistic insight into the events following Bax activation, this observation alone can be considered as a substantial advance. However, there are several issues that should be addressed.

It would be interesting to quantify the 'density' of rings on both fragmented mitochondria and elongated mitochondria following Drp1 knockdown, in order to investigate the importance of Drp1-mediated fission for Bax pore formation.

The two outer membrane proteins (Tom20 and Tom22) which are used to show that the area enclosed by the Bax rings is devoid of proteins, are subunits of the same protein complex. The authors' conclusion would be considerably enhanced if another outer membrane protein that is not part of the TOM complex could also be shown to be excluded from the rings.

Have the authors analyzed Bax immunostaining and ring formation in cells treated with an apoptotic inducer, without addition of ZVAD? Inhibition of caspases may result in non physiological accumulation of Bax in mitochondria and thus formation of structures that may not appear under normal physiological conditions.

It is unclear how the authors can distinguish between lipidic or proteinaceous pores based on their results. Lipidic pores are expected to be unstable. Could this explain why the majority of Bax proteins form aggregates which could correspond to closed pores?

The authors report that fewer Bax rings are visualized on mitochondria from Drp1-deficient cells. This would support previous results according to which Drp1 is required to stimulate Bax oligomerization/pore opening through membrane remodeling (Montessuit et al. 2010). This should be discussed.

The large diameter of the Bax rings is intriguing. I wonder whether these Bax rings surround the entire organelle (similar to Drp1 rings) and rapidly constrict the mitochondria leading to a leaky fission event. Could the authors comment on this possibility?

The results presented in Figure 6 seem to be redundant with those presented in Figure 1D.

Reviewer #1:

Defining the nature and structure of the enigmatic apoptotic pore is one of the most important questions in the field. The current manuscript describes STED super resolution microscopy to attempt to image the putative apoptotic pore. Using immunohistochemistry of Bax in U2OS cells, the authors report the detection of large, bright clusters but also ring structures the lumen of which is devoid of mitochondrial membrane proteins. They also report that the formation of these rings is not sufficient for cytochrome c release in the absence of Drp1-dependent mitochondrial remodelling. The manuscript is very clearly written and the data for the most part convincing and objectively interpreted.

General comments

1. In order to ascertain the generality of their findings, their studies should be extended to multiple cell lines. Also to rule out the possibility that the structures, particularly the clusters are not an artefact of formaldehyde fixation it should be complemented with GFP-Bax expressed in Bak/Bax null cells.

We thank the reviewer for suggesting to analyze multiple cell lines. For the revised version of the manuscript we have analyzed four additional cell lines (HT1080, SHSY5Y, HeLa and CV-1) (see new Fig. 1E). In all four cell lines we were able to demonstrate the formation of Bax-rings on apoptotic mitochondria. In all cell lines analyzed in this paper we find that the area enclosed by the Bax-ring is largely devoid of membrane proteins. Hence we show the phenomenon of Bax pores now in five independent cell lines. This important finding is summarized in Figure 1.

Fixation induced artifacts: As proposed by the reviewer it is indeed very difficult to imagine that a complex structure such as a Bax-ring could be a fixation artefact. Richard Youle and colleagues (JCB, 2001, 153: 1265-76) and others have repeatedly described the occurrence of Bax clusters in living apoptotic cells. We think that these findings together with our observations largely rule out the possibility that these clusters are merely a product of fixation. We agree that live cell imaging of GFP-Bax expressed in Bak/Bax null cells would be an important experiment and it will certainly be done in the future. In fact, we hope that this manuscript will initiate such studies. Nonetheless, there is a potential caveat with GFP-Bax expressed in Bak/Bax null cells: The GFP-tag might have an influence on the ability of Bax to form rings. Hence the absence of Bax-GFP-rings would not necessarily provide evidence that Bax cannot form rings. Thus we are convinced that the presented data on non-tagged Bax is strong evidence for the ring formation. Although we agree that live cell imaging is important in future studies, we think that it is slightly beyond the scope of this manuscript.

2. In Drp1 k/d cells, the authors report other arc-like structures. These could represent partially formed pores or rings comprising other mitochondrial proteins in particular Bak. This should be addressed by examining Bak^{-/-} cells (MEFs or HCT116 for example). Also can the authors show that 100% of Bax is labelled with antibody?

This is also a very good suggestion for an experiment that should be performed in the future. Indeed, we mention the possibility that also other proteins are components of the Bax-rings in the discussion of the revised manuscript:

Page 11: "In addition to smoothly labeled Bax-rings, we repeatedly observed also Bax-rings that were non-homogeneously labeled. This might indicate that at least a sub-set of the Bax-rings are assemblies of Bax with another protein, potentially Bak. Indeed, activated Bax has been reported to interact with Bak into complexes involved in MOMP (Nechushtan et al, 2001; Zhou & Chang, 2008). Hence our data would be in line with ring-like structures consisting of Bax, Bak and potentially other proteins."

We also state explicitly, that we cannot rule out the possibility that not all Bax proteins are labeled (Page 6): "Frequently, the labeling of the rings appeared to be speckled. This might be due to imperfect antibody decoration, or it might indicate that these assemblies are composed of different proteins."

Together, we think that examining Bak-/cells is indeed an important future experiment, but we believe it would not affect the major conclusion of this manuscript, namely that Bax can form pores. Therefore we believe that it would be beyond the scope of the current study.

3. The authors conclude that the clusters are "loosely attached" to mitochondria. This cannot be determined even with super resolution microscopy and should be interpreted more cautiously.

We thank the reviewer to pointing us to this issue. He/she is absolutely right; we have been over-interpreting the data. We interpret it now more cautiously and changed the main text as well as the figure legend to Suppl. Fig. 1.

The key sentence was changed into (page 5): "STED super-resolution microscopy (or nanoscopy) enabling a lateral resolution of better than 40 nm (Hell, 2009) revealed that these Bax clusters often appeared not to be embedded into the outer membrane but to be attached to the mitochondrial surface with only a fraction of the cluster extending in the outer membrane (Suppl. Fig. 1)."

4. The authors correlate the formation of the ring structure with MOMP in apoptotic cells. What percentage of the cells treated with actD at this timepoint were apoptotic? How were individual cells determined to be apoptotic? Also this aspect of the study would be significantly strengthened by imaging mutants of Bax that do not oligomerise and/or mediate MOMP.

Thank you for pointing to the missing information. 3-4 % of all cells attached to the cover glass were apoptotic at this timepoint. We judged cells that showed a translocation of Bax to the fragmented mitochondria as apoptotic.

This is now explicitly stated in the Results section (page 5): "At this time-point in 3-4% of all cells (n=550) Bax was translocated to the mitochondria which was accompanied by mitochondrial fragmentation. These cells were regarded as apoptotic."

Again, we thank the reviewer for suggesting an important experiment that needs to be performed in the future. We are convinced that the data presented in this study support the presented conclusions. As detailed above, we think that the analysis of Bax mutants in Bax null cells is beyond the scope of the study. Nonetheless, the suggested experiment will be very valuable. We hope that our study will trigger others to perform this and other experiments to reveal further mechanistic details of the Bax-rings.

5. That Smac is released indicates that MOMP has occurred but suggests that Drp1 is required for full cytochrome c release from cristae. These cells will likely still die and undergo cytochrome c release just not as quickly. So what happens at later timepoints? The statement should be more explicit to state that these assemblies are not sufficient for FULL cytochrome c release.

Right. We need to be precise here. We have carefully changed this statement throughout the manuscript.

For example, the revised manuscript states (page 9): "The majority of these Bax-positive mitochondria had released Smac/DIABLO whereas the release of cytochrome c was delayed. We identified numerous mitochondria exhibiting Bax-rings that had not fully released cytochrome c, evidencing that in Drp1 knockdown cells these Bax assemblies are not sufficient for the full release of cytochrome c. This delays but does not inhibit the further progression of apoptosis (Estaquier & Arnould, 2007)."

Specific points

1. Abstract should read, "The Bcl-2 family proteins Bax AND BAK ARE essential..."

We now have included this information at a prominent position in the introduction: Page 3: "Both, Bax and Bak are essential for mitochondria mediated apoptosis."

2. The Owen et al 2008 review should be replaced with the original research article from Xiaodong Wang's group.

Absolutely right. Done.

3. The Introduction discusses Bax translocation to mitochondria during apoptosis. To more accurately reflect current understanding the new phenomenon of "retrotranslocation" should be discussed.

Yes, absolutely right. Thank you for pointing us to this omission.

It is now mentioned in the introduction (page 3): "In healthy cells, Bax and Bak are constantly shuttled between the mitochondria and the cytosol. Because of different rates for the retrotranslocation from the mitochondria to the cytosol, Bak resides predominantly in the mitochondrial outer membrane, whereas the majority of the Bax molecules are in the cytosol (Edlich, Banerjee et al., 2011, Schellenberg, Wang et al., 2013, Todt, Cakir et al., 2015). Upon apoptosis, active Bax is not retrotranslocated, as Bax activation blocks shuttling into the cytosol and consequently Bax accumulates on the mitochondria (Edlich et al., 2011, Wolter, Hsu et al., 1997)."

4. The sentence, "A careful adaptation of the imaging conditions and the color tables demonstrates that almost all Bax-positive fragmented mitochondria in U2OS cells exhibit next to Bax clusters also these unexpected assemblies (Fig. 1C)" should be reworded to clarify.

Thank you for pinpointing this badly written sentence.

It is changed into (page 5): "Even in the STED images these Bax assemblies were not always fully obvious, because they were relatively dim compared to the compact and bright Bax clusters (Fig. 1B). Hence we carefully adapted the imaging conditions and the color tables to visualize also the dim structures. We found that almost all Bax-positive fragmented mitochondria in U2OS cells exhibited next to Bax clusters also ring-like structures and other unexpected assemblies (Fig. 1C)."

Referee #2:

The presented study by Grosse et al used super-resolution STED imaging to investigate how specific proteins, involved in permeabilization of mitochondrial outer membranes, form ring-like structures upon induction of apoptosis. The study also gives indications that the ring-like structures surround holes in the membranes, through which larger proteins can be released. The manuscript is well written, the presented results and conclusions are interesting and seem solid and are likely to provide important clues to the understanding of mitochondrial outer membrane permeabilization. I have no essential suggestions for revisions of this manuscript. Apart from some minor matters, it can in my view be published in its present form.

Thank you for this positive view on our manuscript.

Minor points: p.14, second section: Can the authors give some more information on how the ROIs for the variance determination were selected?

Thank you for pointing us to this omission.

We now explain this in the methods section (page 15): "In brief, first for each individual pixel within the raw STED image the local variance of the fluorescence intensity around the respective pixel was determined in round regions of interest (ROIs) with a diameter of 7 pixels (~140 nm)."

p.14-15, STED microscopy: Please add information on what excitation wavelengths that were used. Was a 775nm laser used for the STED beam, and does then that mean Alexa 488 labeled targets were only confocally recorded?

Thank you for pointing us to this unintended omission. The lacking information has been added to the manuscript:

page 16: "In brief, the fluorophores Alexa594 and KK114 were excited at 594 nm and 640 nm, respectively, and STED was performed at 775 nm for both color channels. The dye AlexaFluor488 was excited at 488 nm and recorded confocally."

Referee #3:

Comments on Manuscript by Große et al.

Using super-resolution STED microscopy, Große et al. have identified a novel ring-shaped structure which is formed during apoptosis. These rings are composed of the mitochondrial outer membrane permeabilization (MOMP)-inducing protein Bax. Furthermore, the area enclosed by the rings seems to be devoid of the mitochondrial outer membrane proteins Tom20 and Tom22, which, according to the authors, would support that idea that Bax forms lipidic pores. Using cells in which the membrane fission protein Drp1 is downregulated, they demonstrate that Bax ring assemblies are not sufficient for the release of cytochrome c. The cristae junction organizing complex MICOS seems to be disassembled after Bax activation and before cytochrome c release.

MOMP is considered as the 'point of no return' during apoptosis, and the events leading to this state have been studied extensively in recent years. It remains to be established however, how Bax oligomerization causes the formation of pores in the outer membrane, and what is the exact nature of these complexes. Formation of the Bax ring structures reported by the authors is thus an extremely interesting observation because it could be the first visualization of the Bax-induced pore in cells, and even though there is little mechanistic insight into the events following Bax activation, this observation alone can be considered as a substantial advance. However, there are several issues that should be addressed.

1. It would be interesting to quantify the 'density' of rings on both fragmented mitochondria and elongated mitochondria following Drp1 knockdown, in order to investigate the importance of Drp1-mediated fission for Bax pore formation.

This is indeed a good suggestion. Unfortunately, the uncertainty in such an estimation would be very high: The surface area of the elongated and the fragmented mitochondria is obviously very different, but difficult to determine. However, this value would be needed for a meaningful analysis, but we cannot extract it from the data. Hence we would prefer not to state this value in the manuscript, because due to the large error it may not help to support the main conclusions.

2. The two outer membrane proteins (Tom20 and Tom22) which are used to show that the area enclosed by the Bax rings is devoid of proteins, are subunits of the same protein complex. The authors' conclusion would be considerably enhanced if another outer membrane protein that is not part of the TOM complex could also be shown to be excluded from the rings.

We thank the reviewer for suggesting this experiment. We have now repeated the experiment using an antiserum against Sam50 instead of antisera against Tom20 or Tom22. We find that the area enclosed by the Bax rings is also devoid of Sam50. This important finding is reported in the new Suppl. Fig. 2A. It strongly supports the concept that the Bax-rings indeed represent pores.

3. Have the authors analyzed Bax immunostaining and ring formation in cells treated with an apoptotic inducer, without addition of ZVAD? Inhibition of caspases may result in non physiological accumulation of Bax in mitochondria and thus formation of structures that may not appear under normal physiological conditions.

This is also a very helpful suggestion. In the revised version of the manuscript we demonstrate that Bax-rings are also formed in the absence of ZVAD. This is shown in the new Suppl. Fig. 2B.

4. It is unclear how the authors can distinguish between lipidic or proteinaceous pores based on their results. Lipidic pores are expected to be unstable. Could this explain why the majority of Bax proteins form aggregates which could correspond to closed pores?

The reviewer is right; we should be more careful here. Hence we no longer discuss in the revised manuscript whether the Bax-rings represent lipidic and proteinaceous pores. The abstract now states (page 2): "Together, our super-resolution data provide direct evidence in support of large Bax-delineated pores in the mitochondrial outer membrane as being crucial for Bax mediated MOMP in cells."

5. The authors report that fewer Bax rings are visualized on mitochondria from Drp1-deficient cells. This would support previous results according to which Drp1 is required to stimulate Bax oligomerization/pore opening through membrane remodeling (Montessuit et al. 2010). This should be discussed.

This is indeed an important point.

In the revised manuscript we write (page 9): "Interestingly, and in line with our observation of fewer Bax-rings on Drp1 deficient mitochondria, a previous study had shown that Drp1 stimulates the oligomerization of Bax, probably by formation of Drp1-induced membrane hemifusion intermediates (Montessuit, Somasekharan et al., 2010)."

The large diameter of the Bax rings is intriguing. I wonder whether these Bax rings surround the entire organelle (similar to Drp1 rings) and rapidly constrict the mitochondria leading to a leaky fission event. Could the authors comment on this possibility?

This is an intriguing idea. However, our data neither support nor negate such a hypothesis. Therefore we would prefer not to discuss it here but to await follow-up studies that may not only raise this question but also to provide a substantiated answer to it.

The results presented in Figure 6 seem to be redundant with those presented in Figure 1D.

There is indeed some overlap. But because Fig. 6 summarizes the main findings of this manuscript we think that some redundancy is justified.

2nd Editorial Decision

09 November 2015

Thank you for submitting your revised manuscript for consideration by the EMBO Journal. It has now been seen by two of the original referees again, whose comments are enclosed.

As you can see, the referees appreciate the revised version and support publication here. There are just a few points to sort out before formal acceptance.

Referee# 1 is in agreement with your response, but still finds that analysis needs to be done on a loss of function Bax mutant to fully support the conclusions drawn. I know that addressing this issue requires additional experimental work, but I also think that including this control will strengthen the findings and make it a really excellent paper. As far as I can see this experiment should be doable within a reasonable timeframe. This is the only outstanding issue and it would be nice if it can be addressed in a good way. Lets discuss this issue further and in particular how long you would anticipate to resolve this last issue.

Thank you for the opportunity to consider your work for publication. I look forward to hearing back from you and to your revision.

Referee #1:

The authors have now provided an updated manuscript that more clearly describes the data presented and also now show new data supporting the appearance of Bax ring structures in the mitochondria of 4 other cell types providing evidence that this may be a general phenomenon in most apoptotic cells. That said, I do not find the conclusion that these structures are devoid of other mitochondrial membrane proteins particularly convincing for the images shown particularly for HeLa cells and SH-SY5Y cells. The neuroblastoma line incidentally should be labelled "SH-SY5Y" and not "SHSY-5Y".

I agree with the authors argument that the structures are unlikely to be a fixation artefact and that imaging of GFP-Bax is not necessarily straight forward. I also agree that whether Bak or other mitochondrial proteins are constituents of the pore is discussed in the text and that it is not necessary to address experimentally in the context of the current manuscript. However, I still think that the analysis of loss of function Bax mutants, rather than be beyond the scope of the current manuscript, would be a really important addition and would significantly strengthen the conclusions that these

ring-like structures are not just epiphenomenon.

Specific comments

The first line of the abstract is not correct. As stated in my first review Bax OR Bak are essential for the execution of intrinsic apoptosis. Both the Abstract and not just the Introduction text need to be corrected.

Referee #3:

The revised manuscript has been strengthened and I think the paper is now ready for publication in EMBO J.

additional correspondence - editor

13 November 2015

Thank you for our recent telephone conversation outlining that you don't have Bax-Bak double knock-out cells and that the final revision of your manuscript will take at least three months if addressing all concerns.

I have discussed the specific control requested by referee #1 with my colleagues, as well as with referee #3.

Referee #3 is in agreement with referee #1 that this would be a good control to add.

I have discussed this issue further with my colleagues, and we do appreciate that you provide the correlation between Bax rings and cytochrome c release. While we agree with the referees that it would be good to add the additional control, we also concluded in light of the above experiment that at this stage it will be sufficient to address the remaining concern in the discussion of your manuscript.

I would thus like to ask you to add to the discussion section that the correlation between ring formation and cytochrome c release would be further strengthened by imaging mutants of Bax that do not oligomerize or mediate MOMP.

I am looking forward to receiving the final version of your manuscript.

2nd Revision - authors' response

23 November 2015

Thank you for your positive view on our manuscript and your very constructive suggestions.

As requested by referee #1 we have

i) replaced the images of the HeLa cells and the SH-SY5Y cells in Fig. 1E by new images of the same cell lines. These new images show the lack of mitochondrial proteins within the Bax-rings more convincingly.

ii) corrected the typo "SHSY-5Y" throughout the manuscript. The cell line is now correctly named "SH-SY5Y".

iii) changed the first line of the abstract. It now reads "The Bcl-2 family proteins Bax **and** Bak are essential for the execution of many apoptotic programs."

As suggested by you we have introduced a new paragraph into the discussion:

"Deletion of the C-terminus of Bax abolishes its mitochondrial localization and its pro-apoptotic activity (Nechushtan, Smith et al., 1999, Schinzel, Kaufmann et al., 2004). Also, apoptosis is largely suppressed in cells expressing Bax mutants that localize to mitochondria but are affected in their

ability to homo-oligomerize (Peng, Tong et al., 2013). Some Bax mutants resulting in a compromised Bax retrotranslocation lead to an accumulation of Bax on the mitochondrial outer membrane, but require further stimulation to facilitate apoptosis (Todt et al., 2015, Todt, Cakir et al., 2013). We propose that the investigation of cells expressing such mutants will help to further dissect the function of the Bax-rings in the succession of events leading to cytochrome c release.”

3rd Editorial Decision

01 December 2015

I am pleased to inform you that your manuscript has been accepted for publication in the EMBO Journal.